# Efficacy Studies of a Trivalent Vaccine Containing PCV-2a, PCV-2b Genotypes and *Mycoplasma hyopneumoniae* When Administered at 3 Days of Age and 3 Weeks Later against Porcine Circovirus 2 (PCV-2) Infection

**DOI:** 10.3390/vaccines10081234

**Published:** 2022-08-01

**Authors:** Patricia Pleguezuelos, Marina Sibila, Raúl Cuadrado-Matías, Rosa López-Jiménez, Diego Pérez, Eva Huerta, Mónica Pérez, Florencia Correa-Fiz, José Carlos Mancera-Gracia, Lucas P. Taylor, Stasia Borowski, Gillian Saunders, Joaquim Segalés, Sergio López-Soria, Mònica Balasch

**Affiliations:** 1Centre de Recerca en Sanitat Animal (IRTA-CReSA), Campus de la Universitat Autònoma de Barcelona, 08193 Bellaterra, Barcelona, Spain; marina.sibila@irta.cat (M.S.); raul.cuadrado@uclm.es (R.C.-M.); rosa.lopez@irta.cat (R.L.-J.); diego.perez@irta.cat (D.P.); eva.huerta@irta.cat (E.H.); monica.perez@irta.cat (M.P.); flor.correa@irta.cat (F.C.-F.); sergio.lopez@irta.cat (S.L.-S.); 2OIE Collaborating Centre for the Research and Control of Emerging and Re-Emerging Swine Diseases in Europe (IRTA-CReSA), 08193 Bellaterra, Barcelona, Spain; joaquim.segales@irta.cat; 3Zoetis Belgium S.A., 20 Mercuriusstraat, 1930 Zaventem, Belgium; pepe.mancera@zoetis.com (J.C.M.-G.); stasia.borowski@zoetis.com (S.B.); gillian.saunders@zoetis.com (G.S.); 4Zoetis Inc., 333 Portage Street 300-504SW, Kalamazoo, MI 49007, USA; lucas.p.taylor@zoetis.com; 5Departament de Sanitat i Anatomia Animals, Facultat de Veterinària, Universitat Autònoma de Barcelona, 08193 Bellaterra, Barcelona, Spain; 6Zoetis Manufacturing & Research Spain S.L., Ctra Camprodon s/n Finca “La Riba”, 17813 Vall de Bianya, Girona, Spain; monica.balasch@zoetis.com

**Keywords:** swine, porcine circovirus 2, porcine circovirus 2-systemic disease, *Mycoplasma hyopneumoniae*, porcine respiratory disease complex, trivalent vaccine, efficacy

## Abstract

Four studies under preclinical and clinical conditions were performed to evaluate the efficacy of a new trivalent vaccine against Porcine circovirus 2 (PCV-2) infection. The product contained inactivated PCV-1/PCV-2a (cPCV-2a) and PCV-1/PCV-2b (cPCV-2b) chimeras, plus *M. hyopneumoniae* inactivated cell-free antigens, which was administered to piglets in a two-dose regime at 3 days of age and 3 weeks later. The overall results of preclinical and clinical studies show a significant reduction in PCV-2 viraemia and faecal excretion, and lower histopathological lymphoid lesions and PCV-2 immunohistochemistry scores in vaccinated pigs when compared to non-vaccinated ones. Furthermore, in field trial A, a statistically significant reduction in the incidence of PCV-2-subclinical infection, an increase in body weight from 16 weeks of age to slaughterhouse and an average daily weight gain over the whole period (from 3 days of age to slaughterhouse) was detected in the vaccinated group when compared to the non-vaccinated one. Circulation of PCV-2a in field trial A, and PCV-2b plus PCV-2d in field trial B was confirmed by virus sequencing. In conclusion, a double immunization with a cPCV-2a/cPCV-2b/*M. hyopneumoniae* vaccine was efficacious against PCV-2 infection by reducing the number of histopathological lymphoid lesions and PCV-2 detection in tissues, serum, and faeces, as well as reducing losses in productive parameters.

## 1. Introduction

Porcine circovirus 2 (PCV-2) is the causative agent of the so-called Porcine Circovirus Diseases (PCVD) [1]. PCVD are composed of four main conditions: PCV-2-systemic disease (PCV-2-SD), PCV-2-subclinical infection (PCV-2-SI), PCV-2-reproductive disease (PCV-2-RD) and porcine dermatitis and nephropathy syndrome (PDNS) [2]. PCV-2 has also been associated with respiratory [3] and enteric [4] diseases, although they have lately been considered as being part of PCV-2-SD [5,6].

Up to now, nine genotypes of PCV-2 have been proposed (PCV-2a to PCV-2i) [7,8]. PCV-2a, PCV-2b and PCV-2d are distributed worldwide while the other genotypes are sporadically detected [9]. PCV-2a was the most prevalent genotype during the 1990s, until PCV-2b became predominant around 2000 and, since 2014–15, PCV-2d has become the predominant PCV-2 genotype in North America and Europe [10,11].

Vaccination is a very successful and efficacious tool in controlling PCV-2 infections, and there are numerous commercial PCV-2 vaccines available worldwide [12]. In Europe, all of them are based on inactivated virus or recombinant subunits based on PCV-2a alone or a combination of PCV-2a and PCV-2b [10,13,14], being in some cases combined with a *Mycoplasma hyopneumoniae* (*M. hyopneumoniae)* bacterin. This combined vaccine strategy is frequently preferred as it reduces pig stress and decreases labour cost [15].

The success of commercial PCV-2 vaccines has been endorsed by a reduction in mortality and cull rates in PCV-2-SD [16], a significant increase in the average daily weight gain (ADWG) and a reduction in the frequency of co-infections [17,18,19,20,21]. Moreover, a reduction in PCV-2 viraemia and lymphoid lesions caused by PCV-2 infection has been also demonstrated [14,17,19]. In the case of a PCV-2-SI scenario, several field trials have demonstrated that PCV-2 piglet vaccination is able to improve the following production parameters: ADWG, percentage of runts, body condition and carcass weight [22].

PCV-2 vaccine efficacy in piglets has also been demonstrated in the face of maternally derived antibody (MDA) against PCV-2 [23]. However, the potential interference on vaccine efficacy produced by MDA has not been demonstrated under normal field conditions [24,25,26]. Interestingly, some studies have reported MDA interference with the development of a humoral response after vaccination [14,27,28,29,30], while others have not [17,20].

The present work aimed to elucidate the efficacy of a novel trivalent vaccine containing inactivated Porcine Circovirus 1 (PCV-1)/PCV-2a chimera (cPCV-2a), PCV-1/PCV-2b chimera (cPCV-2b) and *M. hyopneumoniae* bacterin administered in pigs in a two-dose regime at 3 days of age and 3 weeks later. For this purpose, four independent preclinical and clinical studies were performed.

## 2. Materials and Methods

The preclinical and clinical studies presented in this work were performed independently and evaluated by different regulatory agencies. Therefore, some of the analyses performed used different techniques with different lecture windows.

### 2.1. Preclinical Studies

The efficacy of the vaccine was evaluated under preclinical conditions in two different studies, in which seropositive and seronegative pigs were vaccinated and challenged with either PCV-2a or PCV-2b (Table 1). Preclinical studies were approved by the corresponding institutional animal care and use committees (IACUC) from Zoetis and RTI-LCC CRO prior to initiation.

#### 2.1.1. PCV-2a Challenge Study

A total of 90 3-day-old clinically healthy piglets were included in the study. Animals were allotted to their respective treatment group based on sow serological status. Then, at study day (SD) -1, PCV-2 S/P ratio (optical density [OD] of sample/OD of positive control for each Enzyme-Linked ImmunoSorbent Assay [ELISA] plate) values of the piglets were determined and if animals were not seropositive, they were removed from the study. At SD0, one group received an experimental vaccine containing an *M. hyopneumoniae* bacterin adjuvanted with 10% SP Oil as a control product (representing the non-PCV-2 vaccinated group, NV group). The other experimental group tested the investigational veterinary product (V group). The investigational veterinary product (IVP) contained cPCV-2a and cPCV-2b killed viruses and *M. hyopneumoniae* bacterin adjuvanted with 10% SP Oil in a total volume of 1 mL (equivalent to Fostera Gold^®^ and CircoMax Myco^®^). The IVP represented the most-likely herd conditions, in which certain moderate levels of MDA were present at the time of PCV-2 vaccination.

The pigs were intramuscularly (IM) injected with 1 mL of the control or IVP product at 3 and 24 days of age, corresponding to SD0 and SD22, respectively. The challenge time point was determined when the NV mean MDA titre was below S/P 0.2 to ensure that most of the animals were below the ELISA cut-off (0.5 S/P ratio), ensuring maximal susceptibility to viral infection. This value was obtained by SD52, and all pigs were challenged with 4 mL: 2 mL via intranasal (IN) route (1 mL per nostril) and 2 mL IM of a PCV-2a isolate. All pigs were euthanized three weeks after challenge (SD74).

Serum and faecal swabs were collected prior to challenge and twice weekly post-challenge until necropsy and were tested for PCV-2 viraemia and faecal shedding via real-time quantitative PCR (qPCR). PCV-2 antibodies were also measured in serum by ELISA at SD0 and at weekly intervals post-challenge. At necropsy, four lymphoid tissues (tracheobronchial, mesenteric, inguinal lymph nodes and tonsil) were collected from each pig and fixed by immersion in 10% buffered formalin and processed for histopathology and PCV-2 immunohistochemistry (IHC) as indicated in Section 2.3.3.

#### 2.1.2. PCV-2b Challenge Study

The same experimental design indicated above was applied to the preclinical PCV-2b challenge study. Specifically, a total of 69 pigs were randomly allocated into the abovementioned two treatment groups and vaccinated at 3 and 24 days of age.

As mentioned in the previous preclinical study, the challenge time point was determined when the NV mean MDA titre was below S/P 0.2, ensuring maximal susceptibility to viral infection. This value was obtained around SD49/50 and all pigs were then challenged with 2 mL IN (1 mL per nostril) and 2 mL IM of a PCV-2b isolate. All pigs were euthanized three weeks after challenge (around SD71).

Sampling and analyses were performed as described in Section 2.1.1.

#### 2.1.3. PCV-2 Challenge Strains

The PCV-2a isolate 40895, diluted 1:2 in Optimem (Gibco), was used for the PCV-2a challenge. The final stock material had a viral titer of 10^6.0^ TCID_50_/mL.

The PCV-2b isolate FD07, diluted 1:2 in Optimem (Gibco), was used for the PCV-2b challenge. The final challenge material had a viral titer of 10^4.7^ TCID_50_/mL.

### 2.2. Field Trials

#### 2.2.1. Farm Selection

Two different field trials were conducted in two commercial farms located in North-Eastern Spain. The farms were selected based on the existence of problems with PCVD or a history of PCVD in the previous two and a half years.

Farm A was a two-site commercial farm. Sites I + II (breeding and gestation + nursery) had 2660 sows with a weekly farrowing batch system; piglet weaning was performed at approximately 27 days of age. The sow farm was seropositive against *M. hyopneumoniae*, Porcine reproductive and respiratory syndrome virus (PRRSV) and seronegative to Aujeszky’s disease virus (ADV). The gilts and sows were crossbred (Duroc × Landrace). The sow and gilt vaccination farm program included PRRSV, Porcine parvovirus, *Erysipelothrix rhusiopathiae*, Swine Influenza Virus (SIV), *Actinobacillus pleuropneumoniae* and PCV-2 (at weaning, 6 months of age and post-partum) immunizations. At the fattening facilities, pigs were vaccinated twice against ADV.

Farm B was a farrow-to-finish commercial farm with 10,500 sows with a weekly farrowing batch system; piglet weaning was performed at approximately 25 days of age. The sow farm was seropositive against *M. hyopneumoniae* and PRRSV, and seronegative to ADV. The gilts and sows were of the Pietrain breed. The sow and gilt vaccination farm program included immunization against PRRSV, SIV, Porcine parvovirus, *Erysipelothrix rhusiopathiae, Escherichia coli*, *Clostridium perfringens type C*, atrophic rhinitis, ADV, *M. hyopneumoniae* and PCV-2 (at 3 and 6 weeks of age). The gilts were also vaccinated against PCV-2 at 2.5, 6 and 7 months of age. The piglets were vaccinated against PRRSV before weaning and against ADV, PRRSV and SIV at fattening.

#### 2.2.2. Study Design

These clinical studies were blinded, randomized and controlled trials. A total of 3973 male and female pigs (1983 V and 1990 NV) were enrolled in these studies (Table 2).

Animals from Farm A (field trial A) came from three different batches and animals from Farm B (field trial B) came from one single batch. The studied pigs were selected within each batch during the first three days of life and were randomly distributed (blocked by gender) in two groups: vaccinated (V) and non-vaccinated (NV).

The pigs were vaccinated twice (two doses) by IM injection (neck muscle) with CircoMax Myco^®^ (Zoetis Inc., Lincoln, NE, USA) at 2–5 days and at 23–25 days of age. NV pigs received 1 mL of phosphate buffer saline (PBS) IM at each vaccine administration timing. The pigs from each treatment group were housed comingled within the same pens and barns during the study. Males and females were comingled in the maternity and nursery phase, but genders were separated by pen at fattening.

General health observation of the animals was carried out daily throughout the study. Moreover, blood samples from the piglets were collected at six different time points (before first vaccination and at 7, 11, 16, 20 and 25 weeks of age, approximately) for PCV-2 antibody testing by ELISA and to quantify virus levels by qPCR. Faecal swabs were collected at the same time points (but before vaccination) and tested by qPCR. Body weight was recorded 3 times during the study: before the first vaccination, at 16 weeks of age approximately and before going to the slaughterhouse (around 25 weeks of age). The number of animals weighed was not the same at each timepoint due to deviations occurring during the study (death of animals or animals not found at the weighing moment) (Appendix A); therefore, extra animals not selected at the beginning but from the same treatment group were weighed and included in the study.

Dead animals or pigs euthanized for welfare reasons from weaning until the slaughterhouse were necropsied to determine the cause of death. Lymphoid samples (tracheobronchial, mesenteric, superficial inguinal lymph nodes and tonsil) for monitoring PCV-2 associated lesions and antigens were collected at each necropsy and fixed by immersion in 10% buffered formalin and processed for histopathology and PCV-2 IHC as indicated in Section 2.3.3. Moderate to severe histological lesions together with a moderate or high amount of PCV-2 antigens in lymphoid tissues were diagnosed as PCV-2-SD. Pathological analyses were performed in real time, so, when the first PCV-2-SD case was diagnosed, 60 animals (30 animals per treatment group) were randomly selected and necropsied to obtain lymphoid tissues to assess PCV-2 associated lesions and antigen detection by IHQ.

These clinical studies were approved by the Olot Animal Welfare Committee (ID PJ023) and carried out according to the Guidelines on Good Clinical Practices [31].

#### 2.2.3. PCV-2 Genotyping

To ascertain the PCV-2 genotype/s circulating in the farms, Cap gene (ORF2) from 19 serum samples with the highest PCV-2 viral load (6.6–8.3 log_10_ DNA copies/mL) belonging to NV groups was sequenced. Total DNA was extracted from serum samples using the MagMAXTM Pathogen RNA/DNA Kit (Applied Biosystems) following the manufacturer’s instructions. PCV-2 Cap gene was amplified using the primers PCV-2all_F (5′ GGGTCTTTAAGATTAAATYC 3′) and PCV-2all_R (5′ ATGACGTATCCAAGGAG 3′), and the procedure described by Oliver-Ferrando et al. [32] was followed. PCV-2 amplicons were purified with ExoSAP-IT™ (Thermo Fisher Scientific, Vilnius, Lithuania) kit and sequenced by the Sanger method (BigDye^®^ Terminator v3.1 Cycle Sequencing Kit, Foster City, CA, USA) with the ABI PRISM 3130xl Genetic Analyzer (Applied Biosystem^®^, Foster City, CA, USA) at Servei de Genòmica, Universitat Autònoma de Barcelona (Spain). The quality of the sequences was checked using the Finch TV program and trimmed with BioEdit software 7.2.6 (BioEdit, Manchester, UK) [33].

The phylogenetic analysis of the PCV-2 amplicon sequences obtained followed the proposed classification by Franzo and Segalés [7]. The amplicons of the PCV-2 ORF2 gene obtained herein were aligned against the representative strains of the proposed PCV-2 genotypes using MAFFT software [34]. A neighbour-joining method using the p-distance model was used to build the phylogenetic tree with 1000 bootstraps. The phylogenetic tree was further edited using the iTOL software [35] where bootstrap values higher than 70% were maintained.

### 2.3. Laboratory Methods of Preclinical and Field Studies

#### 2.3.1. DNA Extraction and PCV-2 qPCR

DNA from serum and faecal samples collected from preclinical studies were extracted and tested by a non-commercial in-house qPCR following the procedure described by Mancera Gracia, et al. [36]. All Ct values detected were reported as positives and no threshold were applied.

DNA from serum and faecal samples collected from clinical studies was extracted by using the BioSprint 96 DNA Blood Kit following the manufacturer´s instructions. PCV-2 DNA quantification was performed as described in Oliver-Ferrando et al. [37] using a commercial kit (LSI VetMAX Porcine Circovirus Type 2, Life Technologies, Lissieu, France). The limit of detection (LOD) of the technique in serum samples was 4 × 10^3^ DNA copies/mL and in faecal swabs was 1 × 10^4^ DNA copies/mL. The limit of quantification (LOQ) in serum samples and faecal swabs was 1 × 10^4^ DNA copies/mL. qPCR results were log_10_ transformed and interpreted as described below:

Negative results or values below LOD were given a value equal to half of the LOD, this being 3.3 copies/mL for serum samples and 3.7 copies/mL for faecal swabs.

Values between LOD and LOQ were considered positive and were given a value equal to LOQ (4.0 for serum samples and faecal swabs).

Values over LOQ were considered positive and were given the log_10_ qPCR result.

#### 2.3.2. Serology to Detect PCV-2 Antibodies

PCV-2 antibodies from preclinical and clinical studies were detected using a validated in-house PCV-2 antibody ELISA. The in-house ELISA test procedure consisted of a modified indirect ELISA based on recombinant baculovirus-expressed PCV-2 capsid protein [38]. The PCV-2 antigen-coated plate was washed three times using a PBST washing buffer (0.1 M PBS-pH7.2 and 0.3% Tween 20). The sera were diluted 1:6000 in 5% milk diluent, and 100 µL of each diluted serum was incubated with positive and negative antigens at 36 ± 2 °C for 1 h. Excess antibodies were removed by washing 3 times with PBST buffer. Then, 100 µL of diluted peroxidase-labelled anti-pig IgG was added to each well and incubated at 36 ± 2 °C for 1 h. After 3 washings, 100 µL of 3,3′,5,5′ tetramethylbenzidine (TMB) substrate was added and incubated for 20 min at 36 ± 2 °C. The OD value was measured at 650 nm and 490 nm using a microplate reader and their difference per tested serum was reported as the sample/positive control (S/P) ratio (OD sample–OD negative control/OD positive control–OD negative control). Sera samples with S/P ratio values ≥ 0.5 were considered positive.

#### 2.3.3. Histopathology and PCV-2 IHC

Tissue samples collected at each necropsy (tracheobronchial lymph node, mesenteric lymph node, superficial inguinal lymph node and tonsil) were fixed by immersion in 10% buffered formalin. Then, the fixed tissue samples were dehydrated and embedded in paraffin blocks. From each paraffin block, two consecutive 4 µm thick sections were cut. One section was stained with haematoxylin-eosin (HE) stain and examined for lesions compatible with PCV-2, including lymphocyte depletion [LD] and histiocytic replacement [HR]. The other section was processed by IHC for PCV-2 antigen detection. These lymphoid samples were scored for microscopic lesions associated to PCV-2 (LD and HR) and the presence of PCV-2 antigens by IHC [39]. Briefly, LD, HR and the amount of PCV-2 antigen were scored from 0 (no lesions/no staining) to 3 (severe lesions/widespread antigen distribution) for each lymphoid tissue collected.

In clinical studies, any pig that died or was euthanized beyond weaning age was classified as PCV-2-SD or PCV-2-SI, if they complied with the following diagnostic criteria:Presence of at least one of the following clinical signs: wasting, weight loss, paleness of the skin, dyspnoea, diarrhoea, jaundice and/or inguinal superficial lymphadenopathy (only applicable to PCV-2-SD cases).LD and/or HR of lymphoid tissues (PCV-2-SI: LD and HR ≤ 1; PCV-2-SD: LD and HR > 1).PCV-2 in lymphoid tissues (PCV-2-SI: IHC ≤ 1; PCV-2-SD: IHC > 1).

### 2.4. Statistical Analyses

Statistical analyses were carried out using the software SAS/STAT (User’s Version 9.4, or higher, SAS Institute, Cary, NC, USA) for both preclinical and field trials. When needed, a logarithm transformation was applied to the data before statistical analyses were carried out. Comparisons were performed between the treatment groups (V vs. NV) from each field trial.

A generalized linear repeated measures mixed model was performed to analyze the following variables from preclinical and field studies after the corresponding data transformation in each study: sera and faecal qPCR results, serology and body weight. When the mixed model did not converge, Fisher’s Exact test was used for analysis.

Linear functions of the least-squares mean for body weights were used to calculate estimates of the ADWG for each period. Moreover, a Pearson Correlation Coefficient was also calculated to evaluate the correlation between PCV-2 antibodies before vaccination and the ADWG during the whole study.

A generalized linear mixed model was performed to analyze the following variables from preclinical and field studies after the corresponding data transformation in each study: ever positive (detected positive on at least one sampling point) for viraemia/shedding, mortality, LD, HR and IHC results separately, and diagnosis of PCV-2-SD or PCV-SI. When the mixed model did not converge, Fisher’s Exact test was used for analysis.

The MDA effect on seroconversion due to vaccination in piglets from field trials was evaluated by calculating a Pearson Correlation Coefficient for the correlation between PCV-2 antibodies before vaccination and the increase in PCV-2 antibodies at 7 weeks of age (Delta value) after natural logarithm data transformation.

The significance level (α) was set at *p* ≤ 0.05 for all statistical analyses.

## 3. Results

### 3.1. Preclinical Studies

#### 3.1.1. PCV-2a Challenge Study

##### Clinical Evaluation

No clinical signs nor mortality were recorded in any studied group.

##### PCV-2 Antibody Detection

Mean ELISA S/P ratios results obtained during the study are represented in Figure 1A. The least-square mean of the S/P ratio declined from SD-1 to SD22 in all treatment groups. There was a significant difference (*p* < 0.01) in the S/P ratios (NV vs. V groups) on SD43 and SD50. After challenge, S/P ratios in the V group were significantly higher (*p* ≤ 0.05) than the NV group on each of the days tested.

##### PCV-2 Viraemia and Faecal Shedding

All pigs were negative for PCV-2 qPCR prior to challenge. From SD57 through SD74, viraemia in the V group was significantly lower (*p* < 0.01) than in the NV (Figure 1B). In addition, the percentage of ever-viraemic pigs was significantly less (*p* < 0.01) in the V group than in the NV group (Table 3).

All pigs were negative for PCV-2 faecal shedding prior to challenge (data not shown). From SD60 through SD71, faecal shedding in the V group was significantly lower (*p* < 0.01) than in the NV group (Figure 1C). The percentage of ever-faecal-shedding pigs was significantly lower (*p* ≤ 0.01) in the V group than in the NV group (Table 3).

##### PCV-2 Detection in Lymphoid Tissues and Microscopic Lymphoid Lesions

The percentage of pigs with HR (*p* ≤ 0.01) and LD (*p* = 0.01) scores and detection of PCV-2 within lesions by IHC (*p* = 0.01) in the V group was significantly lower than those of the NV group (Table 4).

#### 3.1.2. PCV-2b Challenge Study

##### Clinical Evaluation

No clinical signs nor mortality were recorded in any studied group.

##### PCV-2 Antibody Detection

The mean PCV-2 ELISA S/P ratio results obtained during the study are represented in Figure 2A. All piglets in the NV and V groups were PCV-2 serologically positive (S/P ≥ 0.50) on SD0. Pigs in the V group had significantly higher (*p* ≤ 0.01) PCV-2 ELISA S/P ratios after challenge (SD 56/57–SD 70–72) compared to those from the NV group.

##### Viraemia and Faecal Shedding

All pigs were negative for PCV-2 viraemia prior to challenge. The V group animals had a significantly less viral load in serum (*p* < 0.01) compared to those from the NV group at SD 56/57 to SD 70–72 (Figure 2B). The percentage of pigs ever-viraemic was significantly less (*p* < 0.01) in the V group compared to pigs from the NV group (Table 3).

All pigs were negative for PCV-2 in faecal swabs prior to challenge. The V group had significantly less (*p* ≤ 0.01) PCV-2 faecal shedding than the NV group from SD 56/57 to SD 70–72 (Figure 2C). The percentage of pigs ever-faecal-shedding was significantly lower (*p* = 0.05) in the V group compared to the NV one (Table 3).

##### PCV-2 Detection in Lymphoid Tissues and Microscopic Lymphoid Lesions

The percentage of PCV-2 IHC positive pigs was significantly lower (*p* ≤ 0.01) in the group V compared to the NV one. In addition, there was a significantly lower (*p* = 0.01) percentage of pigs with HR in the V group compared to the NV group (Table 4). No significant differences in LD scores were observed between both groups.

### 3.2. Field Trials A and B

#### 3.2.1. Clinical Evaluation

Body weight results, ADWG and mortality are represented in Table 5.

In field trial A, a significantly higher (*p* ≤ 0.04) body weight was observed in the V group at 16 and 24–27 weeks of age compared to the NV group. Moreover, the ADWG was significantly higher (*p* = 0.02) in the V group animals compared to the NV group ones during the whole study period.

In field trial B, no statistically significant differences in body weight nor in ADWG were detected.

It is worth noting that no significant correlation between PCV-2 ELISA S/P ratios before vaccination and ADWG were detected in the V and NV groups of both field trials.

Moreover, no statistically significant differences were detected in mortality between treatment groups from each field trial.

According to the macroscopic lesions detected in the necropsy of animals from field trial B, the high mortality was likely related to an outbreak of *Streptococcus suis* or *Glaesserella parasuis* (no bacteriological investigations were conducted, but those are the most likely agents for cases of fibrinous polyserositis, fibrinous pericarditis and polyarthritis, as we observed in a significant number of necropsies).

#### 3.2.2. PCV-2 Viraemia

All tested pigs from both trials (*n* = 188) were PCV-2 qPCR negative before vaccination.

In field trial A, significantly lower (*p* ≤ 0.05) viral loads in serum were detected in the V group pigs from 11 to 25 weeks of age compared to the NV group ones. In addition, a significantly lower (*p* ≤ 0.01) percentage of PCV-2 viraemic pigs was detected in the V group animals at 20 and 25 weeks of age compared to the NV group ones (Figure 3A). Regarding field trial B, a statistically significant lower (*p* < 0.01) PCV-2 load in serum was observed in the V group pigs at 16 and 20 weeks of age compared to the NV group pigs (Figure 4A).

In addition, the percentage of pigs ever-viraemic (detected positive at least at one sampling point) in both studies were significantly lower (*p* < 0.05) in the V group than in the NV one (Table 6).

#### 3.2.3. PCV-2 Faecal Shedding

In field trial A, statistically significantly lower (*p* < 0.01) PCV-2 loads in faecal swabs was observed in the V group animals from both studies at 20 and 25 weeks of age compared to the NV group (Figure 3B).

In field trial B, statistically lower (*p* = 0.04) PCV-2 faecal shedding in the V group pigs was also detected at 16 weeks of age compared to the NV group (Figure 4B).

Regarding the percentage of positive faecal swabs detected at least in one sampling point, no statistical differences were detected in any of the two studies between the V group pigs (41/44 [93.2%] and 51/57 [89.5%] from field trials A and B, respectively) and the NV group animals (42/43 [97.7%] and 56/66 [84.8%] from field trials A and B, respectively).

#### 3.2.4. PCV-2 Genotyping

The 19 PCV-2 qPCR-positive samples with the highest viral load (6.6–8.3 log_10_ DNA copies/mL), 10 and 9 from field trials A and B, respectively and belonging to the NV groups from both field trials were sequenced to elucidate the main PCV-2 genotype/s circulating in the farms during the study periods (Appendix A). In field trial A, the PCV-2b genotype was found in 6 out of 10 sera analyzed, while PCV-2d was detected in 2 sera; no sequences were obtained in the 2 other samples. In addition, in field trial B, genotype PCV-2a was found in 8 out of 9 serum samples, and no sequence was obtained in 1 serum sample. One single genotype was found per sequenced serum.

#### 3.2.5. PCV-2 Antibody Detection

No statistically significant differences between treatment groups in mean PCV-2 ELISA S/P ratios before the time of treatment administration were found in both studies.

In field trial A, piglets from the V group showed higher (*p* < 0.05) mean PCV-2 ELISA S/P ratios from 7 until 20 weeks of age compared with those from the NV group (Figure 3C).

In field trial B, the gender had a significant treatment interaction effect on serological results; therefore, treatment comparisons for each gender were performed. The vaccinated female pigs showed higher (*p* ≤ 0.01) PCV-2 ELISA S/P ratios at 16 weeks of age compared to the NV ones. In contrast, the V group male pigs had significantly lower (*p* ≤ 0.05) mean PCV-2 ELISA S/P ratios at 20 and 25 weeks of age compared to the NV group (Figure 4C).

The correlation between PCV-2 ELISA S/P ratios of the V group animals before first immunization and the increase in PCV-2 antibody titres at 7 weeks of age (Delta value) is represented in Figure 5. A significantly (*p* ≤ 0.01) negative correlation between the PCV-2 ELISA S/P ratios at first vaccination timing and 7 weeks of age was detected in the V groups from both field trials, indicating that the higher the MDA at vaccination time, the lower the PCV-2 antibody at 7 weeks of age. No significant correlation was obtained for the NV groups in both field trials (data not shown).

#### 3.2.6. Histopathology and PCV-2 IHC

Table 7 summarizes the histopathology and IHC results of studied lymphoid tissues in dead or euthanized pigs during the field trials’ duration.

In field trial A, the percentage of animals diagnosed as PCV-2-SD was 2.2% (2 out of 91 pigs) in the NV group and 0.0% (0 out of 81 pigs) in the V group. In contrast, a significantly higher (*p* = 0.03) proportion of the NV group animals were diagnosed as PCV-2-SI (20 out of 89 [22.5%]) compared to the V group (8 out of 81 [9.9%]). Regarding pathological findings, the NV group animals had a significantly higher (*p* = 0.02) positive PCV-2 IHC scoring compared to that of the V group ones, but no significant differences for the rest of the variables among both studied groups were found.

In field trial B, no PCV-2-SD was detected in any of the studied animals, and no statistical differences in cases of PCV-2-SI were detected between the V and NV groups (NV: PCV-2-SI was detected in 4 pigs out of 220 dead/euthanized pigs [1.8%] and V: PCV-2-SI was detected in 3 pigs out of 171 dead/euthanized pigs [1.8%]). Regarding histopathological findings, a significantly higher (*p* = 0.01) incidence of LD was detected in the NV group pigs compared to the V group pigs, but no significant differences for the rest of the variables among both studied groups were found.

## 4. Discussion

PCVDs are important diseases in swine production worldwide and, since the last decade, vaccination is the main tool for disease prevention [40], these being the main commercial vaccines available, derived from PCV-2a genotype [40,41]. Although PCV-2 vaccines are responsible for PCVD reduction in pig herds, they do not confer full protection and do not eliminate virus replication and transmission [10]. Due to this, PCV-2 vaccination may induce vaccine-escape variants, causing the overall prevalence of PCV-2 positive herds to become unchangeable [10,40,42], and promoting wild-type strains that can circulate in a less susceptible population [43]. Moreover, it has been shown that PCV-2 monovalent vaccines induce protection against a homologous infection but with a lack of full cross-protection against other PCV-2 genotypes [41,43]. Taken together, a bivalent vaccine containing PCV-2a and PCV-2b genotypes would be an interesting option to protect the population against the most clinically relevant PCV-2 genotypes [44]. Hence, the present work reports the efficacy of the results against PCV-2 infection of a new trivalent vaccine containing inactivated cPCV-2a, cPCV-2b and *M. hyopneumoniae* bacterin, administered in a two-dose regimen. This vaccine is built on a current porcine circovirus, Type 1–Type 2 chimera, inactivated virus and *M. Hyopneumoniae* bacterin product, also known as Suvaxyn Combo Circo + MH RTU, with the addition of a PCV-2b capsid protein utilizing the cPCV-2b construct (CircoMax Myco^®^, Zoetis Inc., Lincoln, NE, USA), to provide protection against emerging strains of PCV-2.

The studies presented here were developed independently, these being the preclinical ones performed in the USA and the clinical ones in the EU. Since the results obtained were evaluated by different regulatory agencies and by the requirement of each of the agencies, the data of qPCR were expressed differently in the preclinical and clinical studies. However, the authors consider that this fact does not alter the interpretation of the overall results since the comparison between the preclinical and clinical studies was not the objective of the present article. Moreover, the fact that two different qPCR techniques were used did not interfere with the global assessment of vaccine efficacy on parameters such as productive, serological, viraemia, virus excretion and pathological parameters.

In field trial A, a statistically significant greater body weight and ADWG was observed. Although these differences on body weight were not statistically significant in field trial B, they show a remarkable tendency for improvement of approximately 0.9 kg live weight at 16 weeks of age and 1.0 kg at slaughter, which is notably important from a financial viewpoint [45]. In addition, no correlation between MDA and ADWG was observed in the V group animals. This result suggests that the ADWG was independent of the MDA present at the time of vaccination, as described in other studies [17,46]. Moreover, the global mortality rate from the field studies were lower in the V group piglets than in the NV group ones, although they were not statistically significant. This could be related to the fact that the V group and NV group animals were comingled in the same pen/room. In such a scenario (not frequent under field conditions), the infectious pressure of non-vaccinated piglets may have hindered the vaccine efficacy [47].

A reduction in the amount of PCV-2 positive cells by IHC and lymphoid tissue lesions in the V group animals from preclinical studies was observed. This reduction was statistically significant for the HR and LD variables of the PCV-2a challenge study and for the HR one of the PCV-2b challenge study (only numerical differences were noticed for the LD of the PCV-2b challenge one). These results were corroborated in the field trials, where the vaccinated pigs with the trivalent vaccine had a lower percentage (although non-significant) of animals with lymphoid tissue lesions (HR and LD), and a significantly lower amount of PCV-2 positive cells by IHC, compared to the NV pigs. Based on these pathological results, the incidences of PCV-2-SD and PCV-2-SI were higher in the NV group, but only significantly for PCV-2-SI. These findings confirm previous studies, which have indicated that vaccination reduces microscopic PCV-2-associated lesions and reduces the amount of PCV-2 antigen [48,49,50,51]. In the case of field trial B, the percentage of animals with LD was significantly higher in the NV group, although no statistically significant differences were observed in PCV-2-SD nor PCV-2-SI. These subtle differences are probably due to the low PCV-2 pressure detected at the time of the study performance, showing no PCVD-compatible clinical signs during this study. However, these results are in agreement with those studies reporting that PCV-2 piglet vaccination is effective despite the PCVD farm status (PCV-2-SD or-PCV-2-SI) [13].

The present work demonstrates the ability of a vaccination in a two-dose regimen to stimulate the development of IgG in the presence of MDA or, more evidently, subsequently after a PCV-2 challenge with different genotypes (in both preclinical studies) or natural infection (in case of field trial A). Such immunization would result in a reduction in the PCV-2 loads in serum, faecal excretion, percentage of PCV-2-viraemic pigs (this stands only for field trial A) and percentages of ever-viraemic pigs. These results are in concordance with several published experimental studies where animals were infected with PCV-2 after vaccination at different ages (5 days of age, 10 days of age, 3 and/or 6–7 weeks of age), where a higher PCV-2 antibody response plus a reduction in PCV-2 viral load [14,15,27,28,49,52,53,54,55] and faecal excretion [14,28] were also observed. As indicated above, no significant differences were detected in the percentage of PCV-2-viraemic pigs at each of the sampling time points in field trial B (although significant differences were detected in ever-viraemic pigs at any time point analysis). This result can be explained by the low PCV-2 natural infection, since in this study no PCV-2-SD was detected and only PCV-2-SI was observed.

MDAs are essential for the neonate’s immune response, and it is also an important component that can have an impact on the success of immunization [56]. In the present work, a PCV-2 antibody response of the vaccine dependent on MDA titres was suggested in the field studies, since a statistically significant negative correlation was detected between PCV-2 IgG antibodies before vaccination and PCV-2 IgG antibody evolution up to 7 weeks of age in the V group animals from both field studies. These results are in line with several studies in which a clear interference of MDA in vaccine efficacy in terms of seroconversion has been shown [25,28,29,46,57]. However, a negative MDA effect on humoral immune response in piglets is suggested to be not related to a negative impact on vaccine efficacy except for those cases where MDA titres are high (≥8 log_2_ IPMA antibodies) [26]. In addition, PCV-2 vaccines induce not only humoral immunity, but also a cellular immune response [13,30]. Therefore, PCVD protection in the absence of a specific serologic response can be due to cellular immunity [58], and consequently, the absence of seroconversion after vaccination in the presence of MDA should not be assessed as any negative indicator for the effectiveness as it has been observed in some studies [25,28,59,60,61].

Different PCV-2 genotypes were detected (PCV-2a, PCV-2b and PCV-2d) in the two commercial farms where field studies were performed. In fact, co-infection of several PCV-2 genotypes in the same farm is not rare [62,63,64]. Although several experimental studies have shown cross-protection between the major genotypes worldwide (PCV-2a, PCV-2b and PCV-2d) [14,40,65,66,67,68], a closer epitopic relationship between PCV-2b and PCV-2d than between PCV-2a and PCV-2d genotypes [10,68,69] has been detected. Although this needs to be demonstrated at an efficacy level, these data may suggest that PCV2b-based vaccines could offer better protection against PCV-2d compared to PCV-2a-based vaccines [70]. In fact, in a recent study by Bandrick et al. [43], animals vaccinated with a cPCV-2a/cPCV-2b bivalent vaccine showed higher levels of protection compared to PCV-2a and PCV-2b monovalent vaccines against PCV-2a and PCV-2b challenges. Animals treated with the bivalent vaccine showed less (although non-significant) PCV-2 shedding in faeces, ever-shed PCV-2 in their faeces, viraemia and ever-viraemic pigs compared to animals treated with the monovalent vaccine. These results are in concordance with the new vaccine used in the present study containing cPCV-2a and cPCV-2b genotypes, therefore expanding the epitopic repertoire of the vaccine product and potentially inducing a wider protection than monovalent vaccines against heterologous PCV-2.

## 5. Conclusions

According to the results of the present preclinical and field studies, a double immunization at 3 days of age and 3 weeks later with the novel trivalent cPCV-2a/cPCV-2b/*M. hyopneumoniae* vaccine was effective against PCV-2 infection by reducing the number of histopathological lymphoid tissue lesions and PCV-2 detection in tissues (IHC), serum and faeces (qPCR), as well as reducing losses in productive parameters (BW and ADWG).

## Figures and Tables

**Figure 1 vaccines-10-01234-f001:**
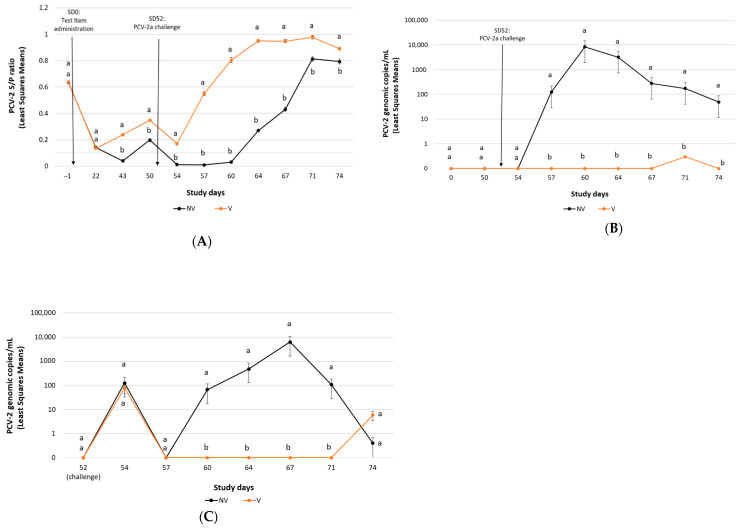
PCV-2a challenge study results: PCV-2 ELISA (mean S/P ratio ± SE) (panel **A**), PCV-2 viraemia load (mean PCV-2 DNA copies/mL ± SE) (panel **B**) and PCV-2 faecal shedding load (mean PCV-2 DNA copies/mL ± SE) (panel **C**). A baseline of 0.1 instead of 0 was used for graphing purposes in (**B**,**C**). Different letters indicate significant differences among experimental groups (*p* ≤ 0.05).

**Figure 2 vaccines-10-01234-f002:**
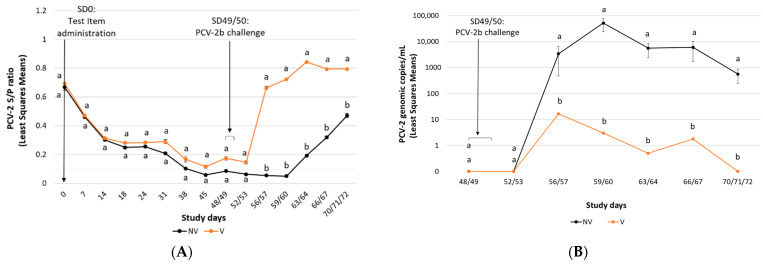
PCV-2b challenge study results: PCV-2 ELISA (mean S/P ratio ± SE) (panel **A**), PCV-2 viraemia load (mean PCV-2 DNA copies/mL ± SE) (panel **B**) and PCV-2 faecal shedding load (mean PCV-2 DNA copies/mL ± SE) (panel **C**). A baseline of 0.1 instead of 0 was used for graphing purposes in Figure 1B,C. Different letters indicate significant differences among experimental groups (*p* ≤ 0.05).

**Figure 3 vaccines-10-01234-f003:**
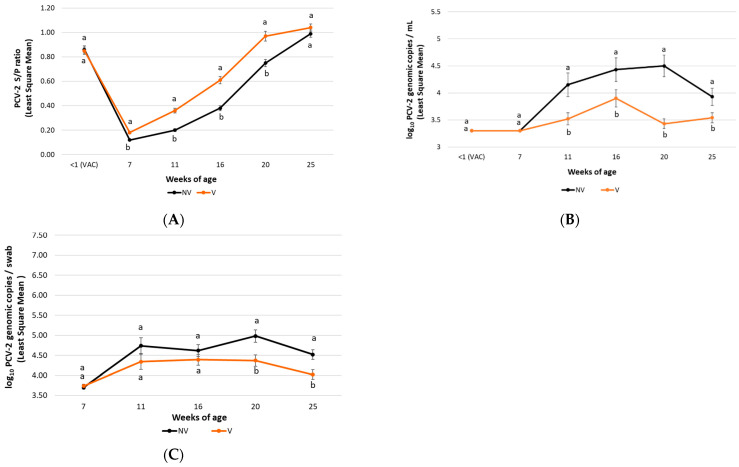
Field trial A results: PCV-2 IgG ELISA S/P results (mean ± SE) in serum samples (panel **A**), PCV-2 viraemia evolution (mean log_10_ genomic copies/mL ± SE) (panel **B**) and PCV-2 qPCR results (mean log_10_ genomic copies/swab ± SE) in faecal samples (panel **C**) at different time points. Different letters indicate significant differences among experimental groups (*p* ≤ 0.05).

**Figure 4 vaccines-10-01234-f004:**
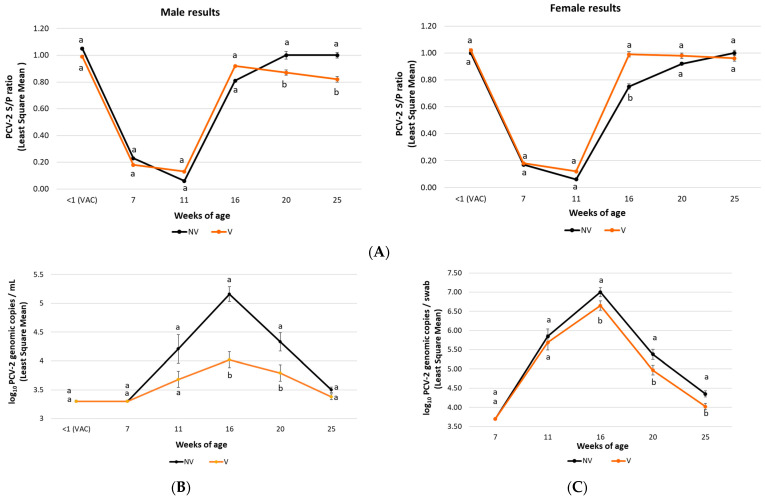
Field trial B results: PCV-2 IgG ELISA S/P results (mean ± SE) in serum samples at different time points (panel **A**). Treatment comparisons for each gender were performed due to a significant treatment interaction effect on serological results. Moreover, PCV-2 viraemia evolution (mean log_10_ genomic copies/mL ± SE) (panel **B**) and PCV-2 qPCR results (mean log_10_ genomic copies/swab ± SE) in faecal samples (panel **C**) at different time points are shown. Different letters indicate significant differences among experimental groups (*p* ≤ 0.05).

**Figure 5 vaccines-10-01234-f005:**
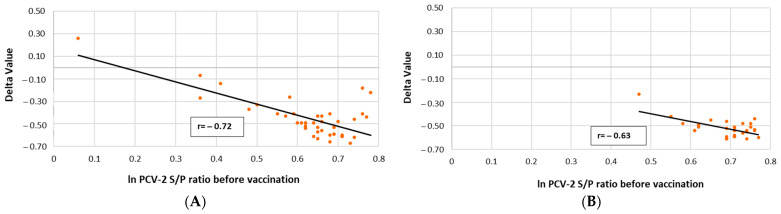
Linear regression and Pearson correlation coefficient between PCV-2 ELISA S/P ratios at vaccination and the increase in these titres until approximately 7 weeks of age (Delta Value) in vaccinated piglets of field trial A (**A)** and field trial B (**B**).

**Table 1 vaccines-10-01234-t001:** Experimental study design and vaccination schedule of pre-clinical studies.

Experimental Groups	Pre-Clinical Studies									
	PCV-2a Challenge					PCV-2b Challenge				
	PCV-2 MDA (Mean S/P Ratio ± SE) at SD0 *	*N* *	NV/V Administration	Challenge	Necropsy	PCV-2 MDA (Mean S/P Ratio ± SE) at SD0	*N* **	NV/V Administration	Challenge	Necropsy
**NV**	0.63 ± 0.01 (seropositive)	45	3 and 24 days old	8 weeks of age approx. (SD52)	10–11 weeks of age (SD74)	0.67 ± 0.02 (seropositive)	34	3 and 24 days old	7–8 weeks of age (SD49-50)	10–11 weeks of age (SD70-72)
**V**	0.64 ± 0.01 (seropositive)	45				0.69 ± 0.02 (seropositive)	35			

NV: Control Product (*M. hyopneumoniae* bacterin only); V: Investigational Veterinary Product; MDA: Maternal derived antibodies; SD: Study day; SE: Standard Error; N: number of animals included. * Thirteen animals from NV group and ten animals from V were removed from the study because they were laid on or due to an unacceptable PCV-2 antibody value in a retrospective piglet serology. ** Seventeen animals from NV group and fifteen animals from V group were removed from the study because their sow was found dead or due to an unacceptable PCV-2 antibody value in a retrospective piglet serology.

**Table 2 vaccines-10-01234-t002:** Experimental study design and vaccination schedule of clinical studies.

Field Trial	Farm	Treatment	Num. of Animals	Doses and Volume	Age at Vaccination
**Field trial A**	**Farm A**	V	1017	2; 1 mL	2–4 and 23–25 days of age
		NV	1021	2; 1 mL	
**Field trial B**	**Farm B**	V	966	2; 1 mL	2–5 and 23–25 days of age
		NV	969	2; 1 mL	

V: Vaccinated (IVP); NV: Non-vaccinated (PBS).

**Table 3 vaccines-10-01234-t003:** Proportion and percentage of ever-PCV-2-viraemic or ever-PCV-2 faecal-shedding pigs in PCV-2a and PCV-2b challenge studies.

Group	PCV-2a Challenge Study		PCV-2b Challenge Study	
	Percentage of Ever Viraemic Pigs	Percentage of Ever Faecal Shedding Pigs	Percentage of Ever Viraemic Pigs	Percentage of Ever Faecal Shedding Pigs
**NV**	31/32 (96.9%) ^a^	30/32 (93.8%) ^a^	17/17 (100.0%) ^a^	17/17 (100.0%) ^a^
**V**	1/35 (2.9%) ^b^	19/35 (54.3%) ^b^	7/20 (35.0) ^b^	15/20 (75.0) ^b^

NV: Control Product (*M. hyopneumoniae* bacterin only); V: Investigational Veterinary Product (IVP). Different letters indicate significant differences among experimental groups NV and V (*p* ≤ 0.05).

**Table 4 vaccines-10-01234-t004:** Histopathology (histiocytic replacement [HR], lymphoid depletion [LD]) and immunohistochemistry [IHC]) results (score > 0) in any of the four lymphoid tissues evaluated (mesenteric lymph node, inguinal superficial lymph node, tracheobronchial lymph node and tonsil) from the PCV-2a and PCV-2b challenge studies.

Group	PCV-2a Challenge Study			PCV-2b Challenge Study		
	HR	LD	IHC	HR	LD	IHC
**NV**	16/32 (50.0%) ^a^	20/32 (62.5%) ^a^	15/32 (46.9%) ^a^	13/17 (76.50%) ^a^	15/17 (88.2%) ^a^	12/17 (70.6%) ^a^
**V**	5/35 (14.3%) ^b^	9/35 (25.7%) ^b^	1/35 (2.9%) ^b^	6/20 (30.0) ^b^	12/20 (60.0%) ^a^	3/20 (1.5%) ^b^

NV: Control Product (*M. hyopneumoniae* bacterin only); V: Investigational Veterinary Product (IVP). Different letters indicate significant differences among experimental groups (*p* ≤ 0.05).

**Table 5 vaccines-10-01234-t005:** Mean body weight (kg ± SE), ADWG (kg/day) and mortality for each field trial. Different letters indicate significant differences among experimental groups (*p* ≤ 0.05) for each field trial.

Study	Group	Body Weight (Kg ± SE)			ADWG (Kg/Day)			Mortality	
		<1 WOA (Vac)	16 WOA	24–27 WOA	<1 WOA to 16 WOA	16 WOA to 24–27 WOA	<1 WOA to 24–27 WOA	Each Treatment Group	Total
**Field trial A**	V	2.2 ± 1.73 ^a^	56.4 ± 1.73 ^a^	114.3 ± 1.73 ^a^	0.47 ^a^	0.90 ^a^	0.63 ^a^	108/896 (12.1%)	221/1801 (12.3%)
	NV	2.1 ± 1.74 ^a^	55.0 ± 1.74 ^b^	112.2 ± 1.73 ^b^	0.46 ^a^	0.89 ^a^	0.62 ^b^	113/905 (12.5%)	
**Field trial B**	V	1.5 ± 0.52 ^a^	45.6 ± 0.48 ^a^	103.4 ± 0.47 ^a^	0.39 ^a^	0.72 ^a^	0.53 ^a^	259/806 (32.1%)	565/1652 (34.2%)
	NV	1.5 ± 0.48 ^a^	44.7 ± 0.45 ^a^	102.4 ± 0.45 ^a^	0.39 ^a^	0.72 ^a^	0.53 ^a^	306/846 (36.2%)	

V: Vaccinated; NV: Non-vaccinated; WOA: Weeks of age.

**Table 6 vaccines-10-01234-t006:** Proportion and percentage of PCV-2 qPCR positive pigs (>3.3 log_10_ DNA copies/mL) at least in one sample point for each experimental group and field trial. Different letters indicate significant differences among experimental groups (*p* ≤ 0.05) for each field trial.

Study	Group	Proportion (%) of Pigs Detected Viraemic Per Sampling Point						Total Proportion (%) of Ever Viraemic Pigs *
		<1 WOA (Vac)	7 WOA	11 WOA	16 WOA	20 WOA	25 WOA	
**Field trial A**	V	0/47 (0.0%) ^a^	0/42 (0.0%) ^a^	7/44 (15.9%) ^a^	22/44 (50.0%) ^a^	12/42 (28.6%) ^a^	3/39 (7.7%) ^a^	30/43 (69.8%) ^a^
	NV	0/50 (0.0%) ^a^	0/44 (0.0%) ^a^	13/43 (30.2%) ^a^	23/41 (56.1%) ^a^	27/39 (69.2%) ^b^	21/40 (52.5%) ^b^	39/43 (90.7%) ^b^
**Field trial B**	V	0/43 (0.0%) ^a^	0/30 (0.0%) ^a^	14/42 (33.3%) ^a^	25/40 (62.5%) ^a^	17/41 (41.5%) ^a^	5/48 (10.4%) ^a^	33/52 (63.5%) ^a^
	NV	0/48 (0.0%) ^a^	0/31 (0.0%) ^a^	15/46 (32.6%) ^a^	42/42 (100%) ^a^	27/37 (73.0%) ^a^	20/53 (37.7%) ^a^	51/65 (78.5%) ^b^

V: Vaccinated; NV: Non-vaccinated; WOA: weeks of age. * Negative animals with a missing value in any of the time points were excluded from the analysis.

**Table 7 vaccines-10-01234-t007:** Proportion of animals with histopathology (histiocytic replacement [HR] and lymphoid depletion [LD]) and immunohistochemistry (IHC) results scores >0 in at least one of the four lymphoid tissues evaluated (mesenteric lymph node, superficial inguinal lymph node, tracheobronchial lymph node and tonsil) corresponding to pigs which died or were euthanized during the study.

Study	Group	HR	LD	IHC
**Field trial A**	**V**	6/81 (7.4%) ^a^	13/81 (16.0%) ^a^	8/81 (9.9%) ^a^
	**NV**	10/91 (11.0%) ^a^	16/91 (17.6%) ^a^	22/91 (24.2%) ^b^
**Field trial B***	**V**	0/172 (0.0%) ^a^	24/171 (14.0%) ^a^	3/192 (1.6%) ^a^
	**NV**	1/220 (1.0%) ^a^	55/221 (24.9%) ^b^	4/241 (1.7%) ^a^

V: Vaccinated; NV: Non-vaccinated. Some tissue samples were not scored by histopathology because the samples were not evaluable. Different letters indicate significant differences among experimental groups (*p* ≤ 0.05) within each field trial.

## Data Availability

Data is contained within the article.

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
