# Peer review of "Efficacy Studies of a Trivalent Vaccine Containing PCV-2a, PCV-2b Genotypes and Mycoplasma hyopneumoniae When Administered at 3 Days of Age and 3 Weeks Later against Porcine Circovirus 2 (PCV-2) Infection"

_vaccines, 2022, doi:10.3390/vaccines10081234_

Round 1

Reviewer 1 Report

           In this manuscript Patricia Pleguezuelos et al. evaluated the efficacy of a trivalent vaccine containing PCV-2a, PCV-2b genotypes and Mycoplasma hyopneumoniae against porcine circovirus 2 (PCV-2) infection in four independent preclinical and clinical studies. Based on their experimental data, the authors concluded that a two-dose immunization regime was effective against PCV-2 infection by reducing the histopathological lymphoid tissue lesions and PCV-2 detection in tissues, sera, and faeces, as well as by reducing losses in productive parameters.

This study has provided important information on the efficacy of the trivalent vaccine against PCV-2 infection. However, there are still a few critical issues that need to be addressed and clarified by the authors.

Major concerns

1.   For “3. Results” section, on lines 303-310, see the Figure on the left.

       Both the vaccinated and the unvaccinated animals were challenged with the PCV-2 isolates. The authors should discuss why at day 74 of the study, the PCV-2 genomic copies per ml went up in the vaccinated animals, whereas the PCV-2 genomic copies per ml went down in the unvaccinated ones.

2.   In Table 5, see lines 444-447. The authors showed that in field trial A, the mortality rate in the vaccinated group was 12.1%, whereas the mortality rate in the unvaccinated group was 12.5%. The authors also showed that in field trial B, the mortality rate in the vaccinated group was 32.1%, whereas the mortality rate in the unvaccinated group was 36.2%.

It seems that there is no substantial difference in the mortality rate between the vaccinated and the unvaccinated groups. The authors claimed that the trivalent vaccine that they employed could reduce infection of PCV-2.

The authors should discuss and mention that the PCV-2 vaccine which they used reduced the rate of infection, however, it did not substantially reduce the mortality rate in the vaccinated group.

3.   The inactivated PCV-2 vaccine was used in this study. The authors should discuss why the traditional inactivated vaccine format was used in the study instead of other vaccine platforms, such as nucleic acid vaccines, or virus vector-based vaccines.

4.    The trivalent PCV-2 vaccine used in this study contains PCV-2a, PCV-2b genotypes, and Mycoplasma hyopneumoniae. The authors should describe why component of Mycoplasma hyopneumoniae was included in the vaccine.

       What is the function of Mycoplasma hyopneumoniae component in the vaccine?

Minor concerns

1. In Materials and Methods section, on lines 241-242, see The PCV-2 antigen-coated plate was washed three times using PBST washing buffer (0.1 M PBS-pH7.2 and 1% Tween 20).

       The final concentration of Tween 20 used for PBST washing buffer is usually 0.1% instead of 1%. The authors should double check whether 1% Tween 20 was used in the experiment.

2.    In Materials and Methods section (2.2.3 PCV-2 genotyping), see lines 198-210.

       The author should briefly describe the conditions for PCR amplification so that other researchers could repeat the experiments without much difficulty.

       The authors should also mention whether the Sanger sequencing was performed on the amplified PCR products or on the purified PCR products.

3.    In “Results” section, on lines 347-348, see “A baseline of 0.1 instead of 0 was used for graphing purposes in figure B and C.

       The words “figure B and C” may be changed to “Figures B and C”.

4.    The authors claimed that the PCV-2d is the predominant PCV-2 genotype in North America and Europe since 2014-2015, why did the authors not include PCV-2d components in the vaccine for the present study?

5.    Whether the vaccinated animals had a reduced rate of co-infection in the current study compared to the unvaccinated ones?

Author Response

Manuscript ID: vaccines-1804025

Manuscript title: Efficacy studies of a trivalent vaccine containing PCV-2a, PCV-2b genotypes and Mycoplasma hyopneumoniae when administered at 3 days of age and 3 weeks later against porcine circovirus 2 (PCV-2) infection

Patricia Pleguezuelos*, Marina Sibila, Raúl Cuadrado-Matías, Rosa López-Jiménez, Diego Pérez, Eva Huerta, Mónica Pérez, Florencia Correa-Fiz, José Carlos Mancera-Gracia, Lucas P. Taylor, Stasia Borowski, Gillian Saunders, Joaquim Segalés, Sergio López-Soria, Mónica Balasch

Dear Editor,

Please find enclosed the revised version (vaccines-1804025) of the abovementioned manuscript addressing the comments from reviewers #1. We are very thankful to all of them for their always constructive criticisms. We really hope that the present version fulfils reviewers’ and editor’s expectations.

As you will see, we have highlighted the changes in the revised version in yellow to facilitate the follow up. We included below the comments of the reviewers and editorial in bold followed by our corresponding answers.

Yours sincerely,

The corresponding author,

Patricia Pleguezuelos

CReSA (Centre de Recerca en Sanitat Animal)

Universitat Autònoma de Barcelona

08193 Bellaterra (Barcelona)

SPAIN

Phone: +34 93 467 40 40 Ext.1730

Fax: +34 93 581 44 90

Response to Reviewer #1 Comments

 In this manuscript Patricia Pleguezuelos et al. evaluated the efficacy of a trivalent vaccine containing PCV-2a, PCV-2b genotypes and Mycoplasma hyopneumoniae against porcine circovirus 2 (PCV-2) infection in four independent preclinical and clinical studies. Based on their experimental data, the authors concluded that a two-dose immunization regime was effective against PCV-2 infection by reducing the histopathological lymphoid tissue lesions and PCV-2 detection in tissues, sera, and faeces, as well as by reducing losses in productive parameters.

This study has provided important information on the efficacy of the trivalent vaccine against PCV-2 infection. However, there are still a few critical issues that need to be addressed and clarified by the authors.

Response: We really appreciate the positive reviewer comments and we tried to address all her/his suggestions as follows.

Major concerns

  1.   For “3. Results” section, on lines 303-310, see the Figure on the left.

Response: Figure 1 has been adjusted in the document.

       Both the vaccinated and the unvaccinated animals were challenged with the PCV-2 isolates. The authors should discuss why at day 74 of the study, the PCV-2 genomic copies per ml went up in the vaccinated animals, whereas the PCV-2 genomic copies per ml went down in the unvaccinated ones.

Response: We appreciate the observation of the reviewer.  In NV group only 1 out of 32 animals was positive and in case of V group 7 out of 35 animals were positives at SD74. These results indicate that very few animals were positive and with low viral load and, because of this, no statistical differences were found. Moreover, there is a technique variability effect that can cause differences at the graph level although, as already mentioned, it was not statistically significant. Therefore, since there are not real differences, we consider it is not needed to highlight this issue.

  1.   In Table 5, see lines 444-447. The authors showed that in field trial A, the mortality rate in the vaccinated group was 12.1%, whereas the mortality rate in the unvaccinated group was 12.5%. The authors also showed that in field trial B, the mortality rate in the vaccinated group was 32.1%, whereas the mortality rate in the unvaccinated group was 36.2%.

It seems that there is no substantial difference in the mortality rate between the vaccinated and the unvaccinated groups. The authors claimed that the trivalent vaccine that they employed could reduce infection of PCV-2.

The authors should discuss and mention that the PCV-2 vaccine which they used reduced the rate of infection, however, it did not substantially reduce the mortality rate in the vaccinated group.

Response: We appreciate the reviewer comment. Mortality parameter is discussed on lines 654-659 indicating that not statistically significance was found. We suggested the fact that V and NV animals being comingled in the same pen/room, which is not usually seen under field conditions, may account for a globally lower infectious pressure of non-vaccinated piglets. However, vaccine efficacy must be evaluated in a wider form, including other clinical aspects such as growth, as well as virological results, antibody response, lesions, etc., and mortality is just one parameter among investigated ones.

  1.   The inactivated PCV-2 vaccine was used in this study. The authors should discuss why the traditional inactivated vaccine format was used in the study instead of other vaccine platforms, such as nucleic acid vaccines, or virus vector-based vaccines.

Response: We really appreciate the reviewer comments. This vaccine derived from a previous commercial product (Porcine Circovirus Vaccine, Type 1 – Type 2 Chimera, Killed Virus, Mycoplasma Hyopneumoniae Bacterin, product known as Suvaxyn Combo Circo + MH in the EU) with the addition of an updated 2b capsid, utilizing the chimeric PCV1-2b construct (CircoMax Myco®), to provide broader protection against emerging strains of PCV2. A clarification has been included on lines 635-638 of the reviewed manuscript. The objective of these studies was to prove the efficacy of this new chimeric vaccine against PCV-2, and no comparisons were planned with other types of vaccines.

  1.    The trivalent PCV-2 vaccine used in this study contains PCV-2a, PCV-2b genotypes, and Mycoplasma hyopneumoniae. The authors should describe why component of Mycoplasma hyopneumoniaewas included in the vaccine.

       What is the function of Mycoplasma hyopneumoniae component in the vaccine?

Response: We appreciate the observation of the reviewer. Mycoplasma hyopneumoniae component is included in the vaccine because Suvaxyn Circo + MH RTU vaccine has been used to produce a novel product adding the chimeric PCV1-2b as mentioned above in a previous comment. This Suvaxyn Circo + MH RTU vaccine already included the M. hyopneumoniae bacterin to protect the pigs against porcine circovirus and M. hyopneumoniae infections. This vaccination strategy has been implemented because vaccination of both pathogens takes place in piglets around the same age as mentioned on lines 58-59 of the reviewed manuscript, and it is very convenient to reduce multiple manipulation of piglets and multiple injections. However, the objective of the present work was to establish the efficacy against PCV-2 specifically and, therefore, the potential efficacy against M. hyopneumoniae was out of the scope of the study. Unlike experimental conditions in which pigs can challenged with both pathogens, it is very difficult to have natural challenge associated with disease presentation with both infectious agents under field conditions.

Minor concerns

  1. In Materials and Methods section, on lines 241-242, see “The PCV-2 antigen-coated plate was washed three times using PBST washing buffer (0.1 M PBS-pH7.2 and 1% Tween 20).”

       The final concentration of Tween 20 used for PBST washing buffer is usually 0.1% instead of 1%. The authors should double check whether 1% Tween 20 was used in the experiment.

Response: In agreement with the comments of the reviewer, percentage of Tween 20 has modified on line 247 of the revised version of the manuscript. It was used Tween 0.3%, instead of 1%.

  1.    In Materials and Methods section (2.2.3 PCV-2 genotyping), see lines 198-210.

       The author should briefly describe the conditions for PCR amplification so that other researchers could repeat the experiments without much difficulty.

Response: We appreciate the reviewer suggestion; however, PCR amplification procedure is already described in Oliver-Ferrando et al. 2016 as indicated on line 210. If I the editor deems it is convenient to include the entire procedure, we will be happy to include it.

       The authors should also mention whether the Sanger sequencing was performed on the amplified PCR products or on the purified PCR products.

Response: Sanger sequencing was performed with the PCR amplicon once purified with ExoSAP-IT™. This information was already stated in lines 210-211 of the revised version of the manuscript.  

  1. In “Results” section, on lines 347-348, see “A baseline of 0.1 instead of 0 was used for graphing purposes in figure B and C.”

Response: Considering that results are log10, baseline 0.1 instead of 0 was used for graphing purpose as indicated on lines 334.

       The words “figure B and C” may be changed to “Figures B and C”.

Response: In agreement with the comments of the reviewer, the change has been included in line 335 of the revised version of the manuscript.

  1. The authors claimed that the PCV-2d is the predominant PCV-2 genotype in North America and Europe since 2014-2015, why did the authors not include PCV-2d components in the vaccine for the present study?

Response: Bandrick et al., 2020 demonstrated that bivalent PCV-2 vaccines have greater T cell epitope overlap with field strains than monovalent PCV-2 vaccines. Moreover, a closer epitopic relationship between PCV-2b and PCV-2d than between PCV-2a and PCV-2d genotypes has been demonstrated. With the objective of updating a PCV-2a based monovalent vaccine (Suvaxyn Circo+MH RTU) to provide a broader immunological coverage, PCV-2b was selected since the combination of PCV-2a and PCV-2b could provide cross-protection for PCV-2d challenge, based on T cell epitope content comparison (EpiCC) analysis.

  1. Whether the vaccinated animals had a reduced rate of co-infection in the current study compared to the unvaccinated ones?

Response: We appreciate the observation of the reviewer. This claim has not been included in the study plan; therefore, it is not possible to answer this question with the results obtained.

Reviewer 2 Report

In this manuscript, the authors assessed the efficacy of the a new trivalent PCV2 vaccine(contained inactivated PCV-26 1/PCV-2a (cPCV-2a) and PCV-1/PCV-2b (cPCV-2b) chimeras, plus a M. hyopneumoniae inactivated cell-free antigens)against PCV-2 infection. Four studies under preclinical and clinical conditions were performed and different PCV2 strains were used as challenged virus.

A total of 3,973 male and female pigs were enrolled in clinical studies. Representative data, including clinical evaluation, ADWG, antibody detection, lymphoid tissues and microscopic lymphoid lesions, and viral loads were analyzed and compared between vaccinated and non-vaccinated infected groups.

The results indicated that piglets vaccinated with this new trivalent vaccine shown vaccine was effective against PCV-2a and PCV-2b infection by reducing the histopathological lymphoid tissue lesions and PCV-2 detection in tissues (IHC), serum and faeces (qPCR), as well as reducing losses in productive parameters (BW and ADWG). Overall, the paper is well written, with clear structures. The data presented also support the conclusion stated.

However, the followings points should be addressed:

1. Line 718: corresponding references should be added here.

 2. For part 3.2.4, the phylogenetic tree analysis of separated PCV2 is carried out. Please supplement the phylogenetic tree diagram to make it more intuitive.

 3. Consider vaccinating piglets at 3 days of age and the presence of maternal antibodies, it is better to introduce the relationship between Maternal derived antibodies and PCV2 vaccine in the introduction.

 4. For part 3.2.1, field trial A and B selected 1017, 1021 and 966,969 piglets respectively for the test, why did they become 896,905 and 806,867 when the survival rate was counted? The mortality rate of piglets in field trial B is more than 30%. It will be more meaningful and interesting to analyze and explore what causes the high mortality rate, whether it is caused by PCV2 or other diseases, and whether the reason for the unsatisfactory immune effect of field trial B is related to the existence of other diseases such as PRRSV, which interferes with the immune effect of PCV2 vaccine.

 5. For part 3.1.1.1, the format and position of pictures and text needs to be adjusted.

 6. Since it is a trivalent vaccine containing M.ycoplasma bacterin, is there any relevant index to detect the immune effect of Mycoplasma? When all components are effective, the vaccine can be called a qualified trivalent vaccine. Whether the part of Mycoplasma antigen has also played an effective role in the immune process?If there are relevant results, it can be supplemented to make the article more perfect.

Author Response

Manuscript ID: vaccines-1804025

Manuscript title: Efficacy studies of a trivalent vaccine containing PCV-2a, PCV-2b genotypes and Mycoplasma hyopneumoniae when administered at 3 days of age and 3 weeks later against porcine circovirus 2 (PCV-2) infection

Patricia Pleguezuelos*, Marina Sibila, Raúl Cuadrado-Matías, Rosa López-Jiménez, Diego Pérez, Eva Huerta, Mónica Pérez, Florencia Correa-Fiz, José Carlos Mancera-Gracia, Lucas P. Taylor, Stasia Borowski, Gillian Saunders, Joaquim Segalés, Sergio López-Soria, Mónica Balasch

Dear Editor,

Please find enclosed the revised version (vaccines-1804025) of the abovementioned manuscript addressing the comments from reviewers #1 and #2. We are very thankful to all of them for their always constructive criticisms. We really hope that the present version fulfils reviewers’ and editor’s expectations.

As you will see, we have highlighted the changes in the revised version in yellow to facilitate the follow up. We included below the comments of the reviewers and editorial in bold followed by our corresponding answers.

Yours sincerely,

The corresponding author,

Patricia Pleguezuelos

CReSA (Centre de Recerca en Sanitat Animal)

Universitat Autònoma de Barcelona

08193 Bellaterra (Barcelona)

SPAIN

Phone: +34 93 467 40 40 Ext.1730

Fax: +34 93 581 44 90

Response to Reviewer #2 Comments

In this manuscript, the authors assessed the efficacy of the a new trivalent PCV2 vaccine(contained inactivated PCV-26 1/PCV-2a (cPCV-2a) and PCV-1/PCV-2b (cPCV-2b) chimeras, plus a M. hyopneumoniae inactivated cell-free antigens)against PCV-2 infection. Four studies under preclinical and clinical conditions were performed and different PCV2 strains were used as challenged virus.

A total of 3,973 male and female pigs were enrolled in clinical studies. Representative data, including clinical evaluation, ADWG, antibody detection, lymphoid tissues and microscopic lymphoid lesions, and viral loads were analyzed and compared between vaccinated and non-vaccinated infected groups.

The results indicated that piglets vaccinated with this new trivalent vaccine shown vaccine was effective against PCV-2a and PCV-2b infection by reducing the histopathological lymphoid tissue lesions and PCV-2 detection in tissues (IHC), serum and faeces (qPCR), as well as reducing losses in productive parameters (BW and ADWG). Overall, the paper is well written, with clear structures. The data presented also support the conclusion stated.

 However, the followings points should be addressed:

Response: We really appreciate the reviewer comments and we tried to address all her/his suggestions as follows.

  1. Line 718: corresponding references should be added here.

Response: We fully agree with reviewer’s comment and reference in line 703 has been added for sentence 701-702.

  1. For part 3.2.4, the phylogenetic tree analysis of separated PCV2 is carried out. Please supplement the phylogenetic tree diagram to make it more intuitive.

Response: We appreciate the reviewer suggestion, and the phylogenetic tree has been added in supplementary material.

  1. Consider vaccinating piglets at 3 days of age and the presence of maternal antibodies, it is better to introduce the relationship between Maternal derived antibodies and PCV2 vaccine in the introduction.

Response: We appreciate the reviewer suggestion, and an introduction of MDA and PCV-2 vaccine has been included on lines 67-71 of the revised version of the manuscript.

  1. For part 3.2.1,field trial A and B selected 1017, 1021 and 966,969 piglets respectively for the test, why did they become 896,905 and 806,867 when the survival rate was counted? The mortality rate of piglets in field trial B is more than 30%. It will be more meaningful and interesting to analyze and explore what causes the high mortality rate, whether it is caused by PCV2 or other diseases, and whether the reason for the unsatisfactory immune effect of field trial B is related to the existence of other diseases such as PRRSV, which interferes with the immune effect of PCV2 vaccine.

Response: We appreciate the observation of the reviewer. Considering the number of animals included in the study, some of them were missed (ear-tag lose mainly).

Moreover, in field trial B, macroscopic lesions associated (fibrinous polyserositis, fibrinous pericarditis and polyarthritis) to Streptococcus suis or Glaesserella parasuis were observed in a significant number of necropsies. Therefore, the high mortality rate observed in this trial was likely related to these two pathogens. A clarification has been included on lines 424-428 of the revised version of the manuscript.

Regarding co-infection, this claim has not been included in the study plan as mentioned above in a previous comment from Reviewer #1; hence, it is not possible to answer this question with the results obtained.

  1. For part 3.1.1.1, the format and position of pictures and text needs to be adjusted.

Response: In agreement with the comments of the reviewer, pictures have been adjusted.

  1. Since it is a trivalent vaccine containing M.ycoplasma bacterin, is there any relevant index to detect the immune effect of Mycoplasma? When all components are effective, the vaccine can be called a qualified trivalent vaccine. Whether the part of Mycoplasma antigen has also played an effective role in the immune process?If there are relevant results, it can be supplemented to make the article more perfect

Response: We appreciate the reviewer suggestion, however, efficacy of Mycoplasma hyopneumoniae was not the objective of this assay. These studies were specifically focused on the efficacy against PCV-2, despite the vaccine is a trivalent product including M. hyopneumoniae antigens. Moreover, to demonstrate the field efficacy of the Mycoplasma hyopneumoniae fraction a field exposure to the agent would have been needed. In these field studies no circulation of Mycoplasma hyopneumoniae was detected, as indicated by absence of seroconversion to Mycoplasma hyopneumoniae at different time points (data not shown).

Round 2

Reviewer 1 Report

No